# Thermal spectrometer for superconducting circuits

Christoforus Dimas Satrya [1] ✉, Yu-Cheng Chang [1], Aleksandr S. Strelnikov [1], Rishabh Upadhyay [1,2], Ilari K. Mäkinen[1], Joonas T. Peltonen [1], Bayan Karimi [1,3] & Jukka P. Pekola[1]

Superconducting circuits provide a versatile and controllable platform for studies of fundamental quantum phenomena as well as for quantum technology applications. A conventional technique to read out the state of a quantum circuit or to characterize its properties is based on RF measurement schemes. Here we demonstrate a simple DC measurement of a thermal spectrometer to investigate properties of a superconducting circuit, in this proof-of-concept experiment a coplanar waveguide resonator. A fraction of the microwave photons in the resonator is absorbed by an on-chip bolometer, resulting in a measurable temperature rise. By monitoring the DC signal of the thermometer due to this process, we are able to determine the resonance frequency and the lineshape (quality factor) of the resonator. The demonstrated scheme, which is a simple DC measurement, offers a wide frequency band potentially reaching up to 200 GHz, far exceeding that of the typical RF spectrometer. Moreover, the thermal measurement yields a highly frequency independent reference level of the Lorentzian absorption signal. In the low power regime, the measurement is fully calibration-free. Our technique offers an alternative spectrometer for quantum circuits.

Superconducting circuits are widely used for quantum phenomena experiments and quantum technology applications[1–4]. Josephson junction (JJ) and coplanar waveguide (CPW) resonator are the central elements as they provide the realization of a macroscopic artificial atom, a quantum bit (qubit), that can be conveniently controlled, probed, and integrated with other non-superconducting elements. For instance, realization of a thermal reservoir by integration of a dissipative element such as a normal-metal absorber provides an experimental platform for studies of quantum thermodynamics[5,6] and heat management[7–9]. Quantum information[10,11] and metrology[12–15] are the most prominent examples of the technology applications of superconducting circuits.

Detection in superconducting devices is a central key in the characterization and operation of the devices. For example, RF spectroscopy becomes a standard technique to measure qubit energy spectrum and its coherent interaction with microwave photons[16–18]. The study of a resonator and its loss mechanisms relies on the spectroscopy technique, either transmission ($S_{21}$) or reflection ($S_{11}$) measurement[19,20]. Conventional RF spectroscopy, involving amplification of a microwave probe tone at different temperature stages followed by heterodyne detection at room temperature, requires electronics and cryogenic elements such as vector network analyzer (VNA), low-temperature amplifiers, circulators, and superconducting coaxial lines. Precise cabling, grounding, and connection of the lines and components are crucial to avoid any microwave mismatch and leakage, that can result to parasitic resonances and suppression of the desired signals[21].

A bolometer consists of an energy-absorbing element and the associated thermometer detecting temperature changes due to energy input[22,23]. Bolometric detection in a superconducting circuit offers advantages compared to conventional schemes of RF

[1]Department of Applied Physics, Pico group, QTF Centre of Excellence, Aalto University, Aalto, Finland. [2]VTT Technical Research Centre of Finland Ltd, Espoo, Finland. [3]Pritzker School of Molecular Engineering, University of Chicago, Chicago, IL, USA. ✉e-mail: christoforus.satrya@aalto.fi

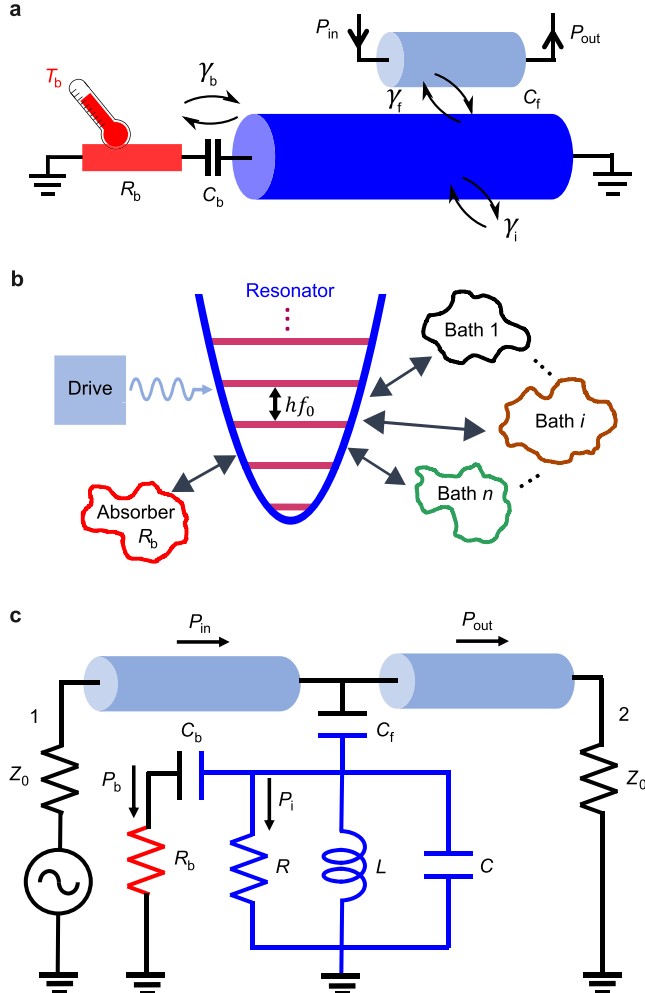

**a**

**b**

**c**

**Fig. 1 | Principle of characterizing a resonator with a bolometer. a** Photons in the resonator (dark blue) are excited due to the input microwave tone $P_{in}$ that is injected to the nearby feedline (light blue). The photons escape via several loss channels, a fraction of them to a nearby capacitively coupled bolometer consisting of an absorber (red) and thermometer. By measuring temperature rise ($T_b$) in the bolometer due to this photon leakage, the total loss rate $2\pi\gamma_t = 2\pi(\gamma_f + \gamma_i + \gamma_b)$ and resonance frequency of the resonator ($f_0$) can be determined. **b** The system is the case of a driven resonator (a harmonic oscillator) coupled to several thermal baths that absorb the photons from the resonator. **c** Equivalent lumped circuit. The bolometer is heated up due to the power $P_b$ transmitted to the absorber $R_b$. The magnitude of $P_b$ depends on the impedance of the other circuit elements $R$, $L$, and $C$. $P_i$ is the power dissipated in the internal bath.

measurement. Using a bolometric detector that can be placed at the millikelvin stage, the use of cryogenic elements can be avoided. A single-shot qubit readout has been demonstrated by using a bolometer that is located on a separate chip[24]. Without the use of a low-temperature parametric amplifier, the authors could measure the qubit state by monitoring temperature in the bolometer. Another advantage is the broadband operation of the bolometer, while a typical RF setup is limited to narrow frequency range of 4–8 GHz. A nano-bolometer has been used to calibrate the attenuation of a coaxial line in a broad frequency range[25]. A bolometric measurement of a high frequency Josephson ac current has been demonstrated[26] where the authors could detect ac current up to 100 GHz. Moreover, the bolometer could reach a high energy resolution and therefore can measure small energies as demonstrated by proximitized thermometer and graphene as the thermal detector[27,28], and it is a potential candidate of a single photon detector.

In this work we design, build, and measure an on-chip bolometer that performs as a sensitive and broadband spectrometer to characterize the properties of a superconducting resonator. The bolometer absorbs the decaying photons from the driven resonator leading to a temperature rise. By measuring the DC signals corresponding to the steady-state temperature in the bolometer, we are able to determine the resonance frequency and the internal quality factor of the resonator. We observe that the loss due to two level systems (TLSs) dominates at the low power regime, while at high power regime quasiparticles (QPs) are the dominant source of losses. The measurements are done by using a simple DC voltmeter.

## Results and discussion

### Principle of thermal spectrometer

The principle of characterizing a superconducting resonator by an on-chip bolometer is shown in Fig. 1a. The resonator (dark blue) is coupled to a feedline (light blue) via capacitance $C_f$. The open-circuit end of the resonator (left) is coupled by capacitance $C_b$ to a bolometer that consists of an absorber (red) with resistance $R_b$ and the associated thermometer to detect temperature change due to input energy. A microwave tone $P_{in}$ is injected through the input feedline to excite photons in the resonator. The photons subsequently decay via several loss channels. Internal loss is mostly due to TLSs or QPs depending on power and temperature, and external losses are dominated by the proximity of the feedline and the bolometer itself. Thus, the total loss rate is $2\pi\gamma_t = 2\pi(\gamma_b + \gamma_i + \gamma_f)$, where $2\pi\gamma_i$ is the internal loss rate, $2\pi\gamma_f$ is loss rate to the feedline, and $2\pi\gamma_b$ is loss rate to the bolometer. The system is the case of a driven resonator (a harmonic oscillator) coupled to several thermal baths, as shown in Fig. 1b. The photons that are absorbed by the bolometer heat the resistor with power $P_b$, leading to temperature rise $T_b$. The power $P_b$ takes Lorentzian form as a function of the driving frequency around resonance $f_0$ with the linewidth $\gamma_t$. Due to the linearity of the resonator, $P_b$ is independent on bolometer temperature and takes a form

$$P_b(f) = \frac{1}{2}\frac{\gamma_f\gamma_b}{(f - f_0)^2 + (\gamma_t/2)^2}P_{in}, \tag{1}$$

a result that one can obtain both by circuit theory and open quantum system approach (see Supplementary Information I and II). By measuring the frequency dependence of $P_b$, which can be measured by a steady-state temperature measurement of the bolometer, the total-loss rate $2\pi\gamma_t$ and the resonance frequency $f_0$ of the resonator can thus be determined.

Moreover, from the lumped circuit model, the heating power $P_b$ depends on the impedance of the circuit elements[29]. At frequency around resonance $f_0$, the resonator can be approximated as a parallel LCR circuit, which is capacitively coupled to the absorber $R_b$ and two ports of feedline $Z_0$ as shown in Fig. 1c (see Supplementary Information I). The driving field $P_{in}$ excites the circuit and the energy losses to the dissipative elements, which is to the feedline $Z_0$, to the bolometer $R_b$, and to the internal bath $R$. The $P_i$ is power dissipated at internal bath. Thus, the total inverse quality factor of the circuit (resonator) is $1/Q_t = 1/Q_i + 1/Q_f + 1/Q_b$, where $Q_t = f_0/\gamma_t$, $Q_f = (2Z_{LC}/Z_0)(C/C_f)^2$, $Q_b = (Z_{LC}/R_b)(C/C_b)^2$, and $Z_{LC} = \sqrt{L/C}$. Therefore, the inverse internal quality factor can be expressed as

$$\frac{1}{Q_i} = \frac{\gamma_t}{f_0} - (Z_0/2Z_{LC})(C_f/C)^2 - (R_b/Z_{LC})(C_b/C)^2 \tag{2}$$

### Experimental setup and result

The studied device consists of a quarter-wavelength ($\lambda/4$) CPW resonator capacitively probed by the feedline and the on-chip bolometer as shown in the device layout in Fig. 2a. Both resonator and feedline are

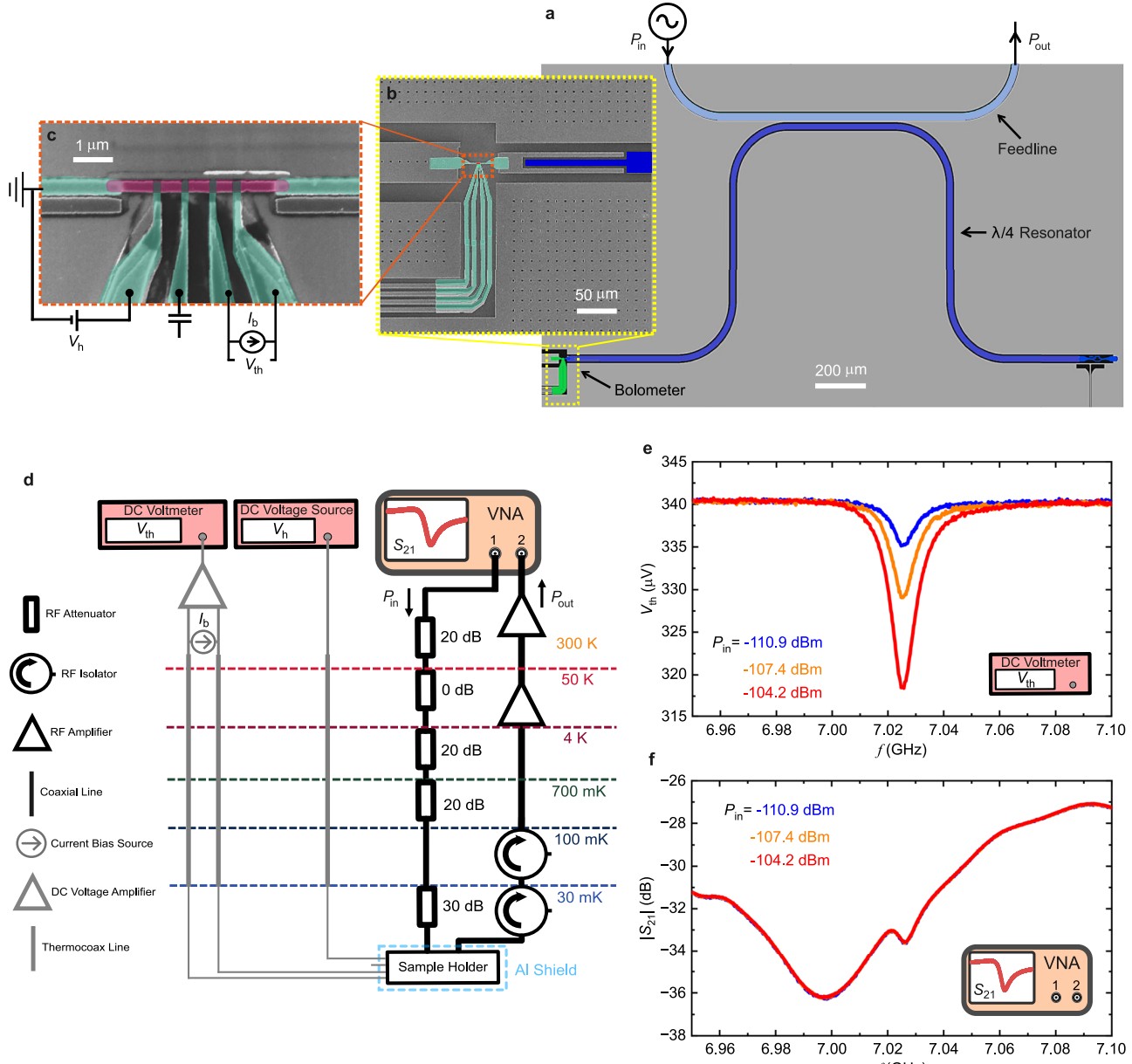

**Fig. 2 | The measured device, experimental setup, and measurement results. a** Layout of the measured device: a $\lambda/4$ CPW resonator (dark blue) is coupled to a probing feedline (light blue) and an on-chip bolometer (inside yellow dashed-line). **b** The open side of the resonator (left side) is weakly capacitively coupled to the bolometer via a clean-contacted interfacing Al film (green). **c** Colored SEM image of the bolometer: Cu film absorber (red) is connected to four Al leads (green) via an insulator forming NIS junctions, for probing the electronic temperature ($T_b$) of Cu metallic island. **d** Experimental setup consisting of DC (left) and RF (right) configuration. **e** Thermometer voltage $V_{th}$ measurements with a DC voltmeter. **f** Scattering measurements $S_{21}$ measured by VNA. The measurements are done simultaneously on the same sample.

made of niobium (Nb) film on top of silicon (Si) substrate with an $AlO_x$ sublayer. We design $C_f \sim 13.85$ fF, $C_b \sim 19.6$ fF, $R_b$ is about $12.23\,\Omega$, equivalent inductance of the resonator is $L = 1.44$ nH, and equivalent capacitance of the resonator is $C = 356$ fF. These values correspond to the quality factor of the feedline and bolometer as $Q_f \sim 1681$ and $Q_b \sim 1716$, respectively. The shorted end of the resonator (right) is shunted by an inactive flux qubit consisting of a parallel aluminum (Al) line and JJs, tuned at zero magnetic flux, thus effectively shunted only by the Al line[30,31]. The open-circuit end of the resonator is capacitively connected to a copper (Cu) film functioning as the bolometer as shown in Fig. 2b. In the experiment, we employ a pair of normal metal-insulator-superconductor (NIS) junctions in SINIS configuration to probe the temperature ($T_b$) in the Cu absorber (red), as shown in Fig. 2c. The temperature rise manifests in the change of voltage $V_{th}$

across the SINIS pair[23,32]. Another single NIS junction is connected to a voltage source ($V_h$) for heating the resistor (increasing $T_b$) to test and calibrate the thermometry. Once the temperature is determined from the calibration conversion (see Supplementary Information III), the power can be calculated by electron phonon relation as

$$P_b = \Sigma\Omega(T_b^5 - T_0^5) - P_e, \qquad (3)$$

where $\Sigma$ is electron phonon coupling constant, $\Omega$ is the volume of the Cu absorber, $T_b$ is the electronic temperature in Cu, $T_0$ is temperature of phonon bath, and $P_e$ is constant background heating from the environment. The volume and electron phonon constant of the Cu absorber are estimated to be $\Omega = 2.52 \times 10^{-20}\,m^3$ and $\Sigma = 2 \times 10^9\,WK^{-5}m^{-3}$ [23,32,33].

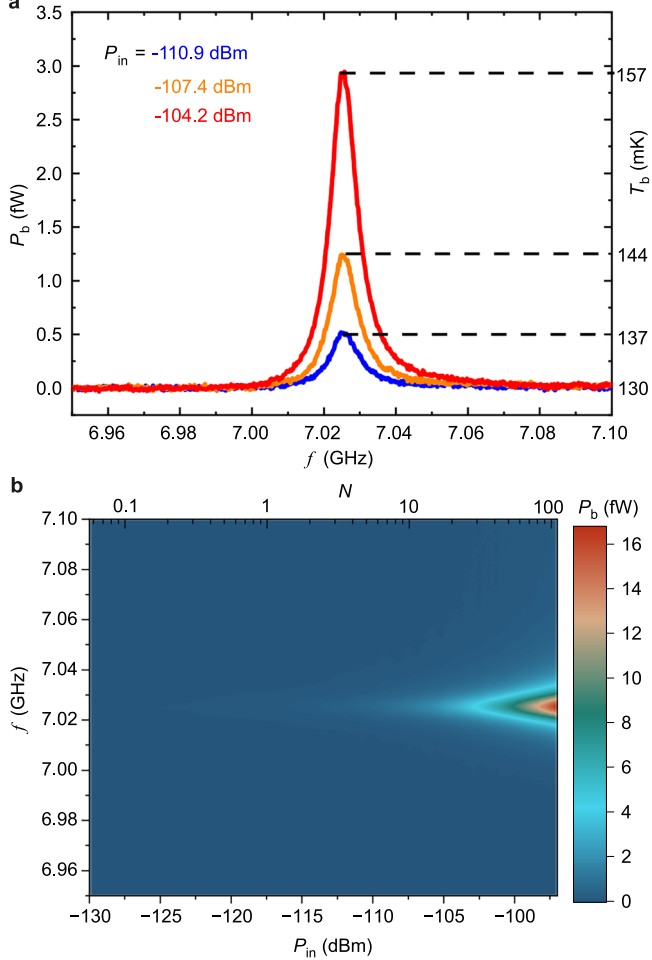

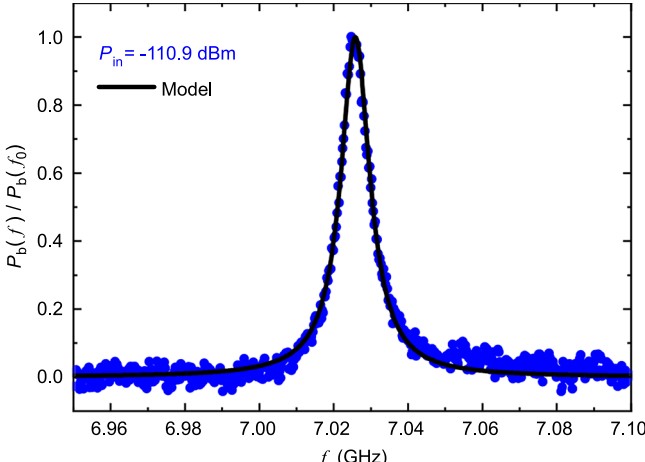

**Fig. 4 | Lorentzian absorption.** Normalized power $P_b(f)/P_b(f_0)$ and fitting with the Lorentzian function of model Eq. (1).

**Fig. 3 | Power measured by bolometer. a** Heating power $P_b$ and bolometer temperature $T_b$ due to photon leakage from the resonator. **b** $P_b$ versus $f$ at different input powers varying from −130 dBm to −95 dBm. Top axis shows the average photon number $N$ in the resonator.

The resonator is probed simultaneously by both the feedline and bolometer. The measurement schemes consisting of DC and RF setup are shown in Fig. 2d. The measurements are performed in a dilution refrigerator at around 50 mK temperature. Input power $P_{in}(f)$ is injected from Port-1 of the feedline, and the output power $P_{out}(f)$ from Port-2 is amplified and measured at room temperature. Comparing the output and input signal by using VNA, the scattering parameter $S_{21}(f)$ can be measured. From the VNA, we sweep the frequency around a resonance frequency at fixed input power varying from −130 dBm to −95 dBm. Simultaneously with the $S_{21}$ measurement, the SINIS voltage $V_{th}$ is recorded with a DC voltmeter. The $V_{th}$ and $S_{21}$ at three different input powers are shown in Fig. 2e, f. We observe noticeable reduction of $V_{th}$ at frequency $f_0 \approx 7.026$ GHz, which is the fundamental resonance frequency. As expected, this voltage reduction is due to heating that increases the temperature in the bolometer, caused by photons emitted by the resonator. The minimum values of $V_{th}$ are located at the resonance frequency where the heating power is at maximum. On the contrary in $S_{21}$ measurement, the resonance dip is weak (-1 dB), buried in the varying background. This is due to the frequency-dependent transmission in the line and the low loaded quality factor of the resonator. From these two measurements, we can see that with the thermal detector, the signal is more pronounced compared to the scattering measurement.

We convert the measured $V_{th}$ to its corresponding temperature $T_b$ and power dissipation $P_b$. By using the conversion formula obtained from the bolometer calibration (see Supplementary Information III), we obtain $T_b$ around the resonance as shown in Fig. 3a on the right axis. Although the measurement is carried out at 50 mK, $T_b$ saturates at around 130 mK. This saturation is dominated by heating from the environment with effective background power $P_e$-2 fW which mostly comes from the DC lines that connect to the NIS junctions[32,34]. The heating power $P_b$, converted from Eq. (3), is plotted in Fig. 3a on the left axis. Figure 3b displays 2d plot of $P_b$ versus $f$ at different powers. At resonance $f_0$, the heating power $P_b$ grows by increasing input power due to the rising of average photon number in the resonator, $N = P_b(f_0)Q_b/2\pi h f_0^2$ (see Supplementary Information IV). The thermal measurement results in a highly frequency-independent reference level of the Lorentzian absorption signal $P_b$.

We normalize the measured $P_b$ by its magnitude at resonance $P_b(f)/P_b(f_0)$ as plotted in Fig. 4 for $P_{in} = -110.9$ dBm. We fit the data with normalized Lorentzian function of model Eq. (1) to obtain the total linewidth $\gamma_t$ as shown in Fig. 4. The plot of $\gamma_t$ and $Q_t$ versus $P_{in}$ is shown in Fig. 5a. It is known that $\gamma_t$ has power dependence that is due to varying internal loss rate $2\pi\gamma_i$. We observe two different behaviors at below and above power around $P_{in} = -102$ dBm. To understand this behavior, we estimate the internal quality factor of the resonator $Q_i$ as plotted in Fig. 5b calculated by Eq. (2). Extraction of $Q_i$ relies on knowledge of circuit parameters. From measured $f_0$, the equivalent capacitance and inductance are determined by $C = 1/8Z_0 f_0$ and $L = 2Z_0/\pi^2 f_0$ (see Supplementary Information I), and the resistance $R_b$ is measured. The $C_b$ and $C_f$ are obtained from simulations. In Fig. 5b, the error bars come from the fitting uncertainty, while the gray area covers total uncertainty due to an additional potential 2% error in $C_b$ and $C_f$. The increase of $Q_i$ at low power regime is well known to be due to saturation of TLSs[19], and at sufficiently high power the resonator is heated up[20]. We confirm this by comparing the data with the model of internal quality factor due to TLSs and QPs.

$$\frac{1}{Q_i} = \frac{1}{Q_{i,\,TLS}} + \frac{1}{Q_{i,\,QPS}} \tag{4}$$

where $Q_{i,\,TLS} = \sqrt{1 + (P/P_c)^{\beta/2}}/\delta_{TLS}^0 \tanh(hf_0/2k_B T_0)$ is due to losses by TLSs[19] and $Q_{i,\,QPS} = Ae^{-P/P_q}$ is phenomenological model describing quasiparticle losses[20]. The list of values of fitting parameters used is shown in Table 1.

Furthermore, we measure the $S_{21}$ and $P_b$ at frequency above 20 GHz to detect the mode of $3f_0$ of the resonator. With the bolometer,

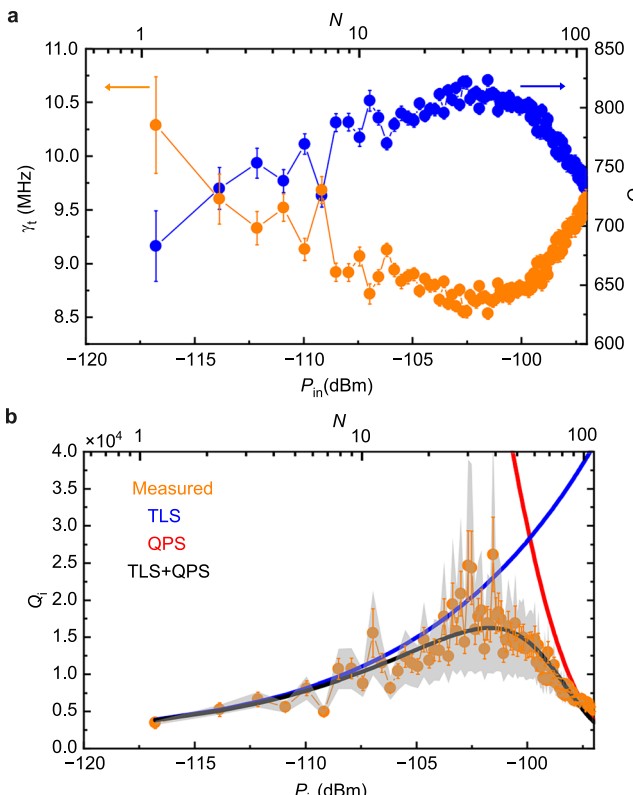

**Fig. 5 | Linewidth and quality factor. a** Total linewidth $\gamma_t$ (orange) and total quality factor $Q_t$ (blue) versus $P_{in}$. The error bars are uncertainty from the fitting. **b** Comparison between measured internal quality factor $Q_i$ (orange) and TLSs $Q_{i,TLS}$ (blue) and QPS $Q_{i,QPS}$ (red) model. The gray area shows total uncertainty due to additional potential error in simulated circuit parameters.

**Table 1 | Fitting parameters for model**

| Quantity | Symbol | Value |
|---|---|---|
| Low power TLSs losses | $\delta_{TLS}^0$ | $5.10^{-4}$ |
| Design-dependent constant | $\beta$ | 2.2 |
| TLSs characteristic power | $P_c$ | −120.79 dBm |
| Resonance frequency | $f_0$ | 7.026 GHz |
| Temperature | $T_0$ | 52.4 mK |
| Amplitude constant | $A$ | $2.3 \times 10^5$ |
| QPs characteristic power | $P_q$ | −103.1 dBm |

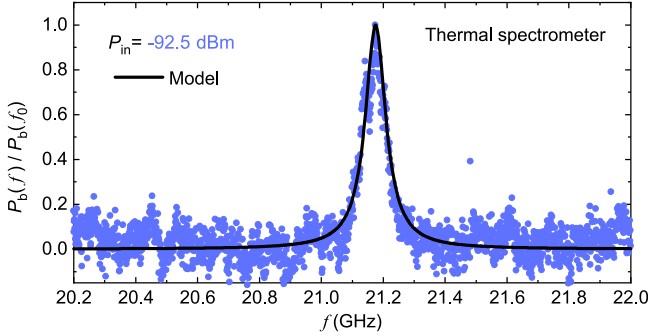

**Fig. 6 | High frequency mode observation, above 20 GHz.** $P_b$ at around resonance of the mode $3f_0$ - 21.2 GHz. With the bolometer, we can clearly observe the Lorentzian power $P_b$ signal around the resonance $3f_0$ with a linewidth -80 MHz. Due to the limitation of the bandwidth of the low-temperature amplifier and isolators, the $S_{21}$ measurement with VNA cannot detect the mode of $3f_0$.

we observe the Lorentzian heating at around frequency $f$ ~ 21.2 GHz, which is the mode of $3f_0$ as shown in Fig. 6. The extracted linewidth is around  -80 MHz. While with the $S_{21}$ scattering measurement, due to the limitation of the bandwidth of the RF low-temperature amplifier and RF isolators, we cannot detect any signal at this frequency range. The bolometer has an estimated cut-off frequency of about 200 GHz, limited by the LR-circuit resonance cutoff $f_c = R_b/2\pi L_b$, where $L_b$ - 8.5 pH is the inductance of the Cu wire. This is a clear demonstration of the advantage of the bolometer thanks to its broad operational frequency compared to standard RF measurement scheme.

Moreover, there is a regime where the thermal spectrometer is fully calibration-free. When the temperature difference $\Delta T = T_b - T_0$ is small ($|\Delta T/T_0| \ll 1$), we can approximate Eq. (3) by $P_b \approx 5\Sigma \Omega T_0^4 \Delta T$[35]. Since in this regime $T_b$ - $V_{th}$ is approximately linear, we have $P_b$ ~ $V_{th}$. Therefore, $V_{th}$ follows already the Lorentzian spectrum. In this case, we can fit directly $V_{th}$ with Lorentzian function without the need to convert it to $P_b$ as shown in Fig. 7a. Figure 7b displays $\gamma_t$ obtained from fitting $P_b$ and $V_{th}$. They show equal results at low power, nearby single-photon regime, but deviate at higher power levels due to large temperature rise, i.e., non-linearity in Eq. (3).

The noise-equivalent power (NEP) extracted from this experiment is $1.4 \times 10^{-18}$ W/$\sqrt{Hz}$. On the other hand, the thermal fluctuation limit is given by $NEP_{th} = \sqrt{4k_B T_b^2 G_{th}} = 2.6 \times 10^{-19}$ W/$\sqrt{Hz}$ at $T_b = 130$ mK, which is the saturation temperature in the current experiment. Thus our experiment is less than one order of magnitude above this fundamental lower bound. For implementation as a qubit readout device, we may compare the experimental performance with the single

photon power $P_b = \hbar f_0^2 2\pi/Q_b = 11.9 \times 10^{-17}$ W at 7.026 GHz with the loss rate $2\pi f_0/Q_b$ from the resonator to the copper strip. This power could be measured using steady-state spectroscopy, even at a relatively high temperature, with the current signal-to-noise ratio at a low speed (several Hz). The thermal spectrometer could then be used in the single-photon regime to perform both one-tone and two-tone spectroscopy of a qubit. This would be useful especially for resonators and qubits operating at high frequency above the standard spectrometer range[36,37]. To operate the bolometer in a fast readout (1 μs resolution) the NIS junctions can be integrated to a LC resonant circuit with intermediate frequency RF setups (-625 MHz) for probing[38]. The measurement time is limited by thermal relaxation which is $\tau = C_h/G_{th}$. The $C_h$ is electronic heat capacity of Cu and $G_{th}$ is thermal conductance to phonon, where $C_h = (71\, J\, K^{-2} m^{-3})\Omega T_b$ and $G_{th} = 5\Sigma \Omega T_0^4$. With $T_b = 130$ mK we estimate the relaxation time to be $\tau = 148$ μs.

In summary, we have demonstrated the operation of an on-chip bolometric spectrometer to characterize a superconducting resonator. The measurement is done by a simple DC measurement setup. The bolometer operates by absorbing the energy decay of the resonator, leading to temperature rise in the bolometer, which is detected by measuring the DC voltage change $V_{th}$ across a pair of NIS thermometer junctions. The resonance frequency and the lineshape (quality factor) of the resonator can be determined based on the temperature change. We have demonstrated additional advantages of the bolometer compared to standard RF measurement in the high-frequency range: the bandwidth of the bolometer exceeds that of a standard RF spectrometer. Furthermore, we found a calibration-free regime where the measured voltage $V_{th}$ follows Lorentzian spectrum. The thermal spectrometer could then be used in the single-photon

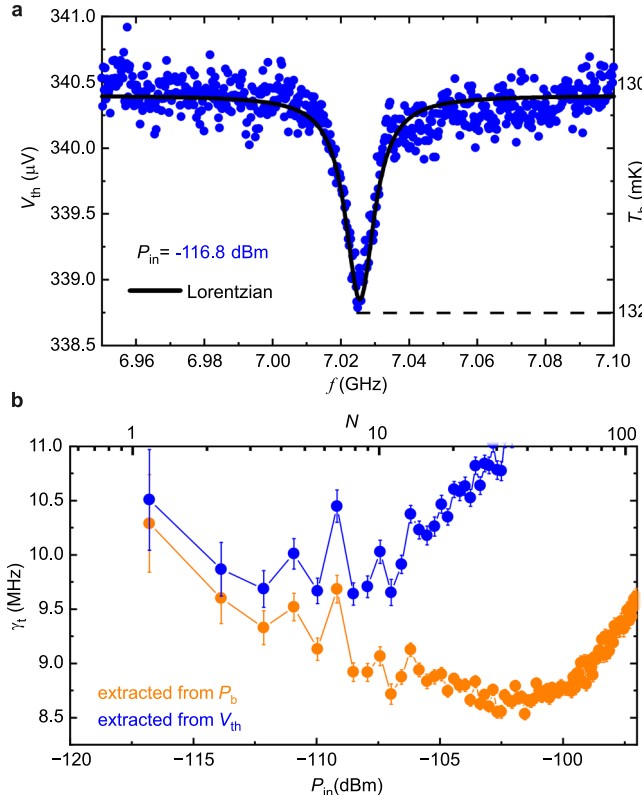

**Fig. 7 | Calibration-free regime. a** At low power, we can fit directly the thermometer voltage $V_{th}$ with a Lorentzian function. **b** Comparison of measured $\gamma_t$ obtained from $V_{th}$ and $P_b$. The error bars are uncertainty from the fitting.

regime to perform both one-tone and two-tone spectroscopy of a qubit (see Supplementary Information V for a preliminary result).

## Methods

### Fabrication

The fabrication of the device is done in a multistage process on a 675 μm-thick and highly resistive Si substrate, resulting to device shown in Fig. 2a–c. The fabrication consists of three main steps: (1) fabricating Nb structures (resonator, feedline, ground plane, and pads), (2) fabricating flux qubit made of three junctions of superconductor-insulator-superconductor (SIS) with aluminum (Al) film, (3) fabricating bolometer consisting absorber and thermometer. A 40 nm-thick $AlO_x$ layer is deposited onto a Si substrate using atomic layer deposition, followed by a deposition of a 200 nm-thick Nb film using DC magnetron sputtering. Positive electron beam resist, AR-P6200.13, is spin-coated with a speed of 6000 rpm for 60 s, and is post-baked for 9 min at 160 °C, which is then patterned by electron beam lithography (EBL) and etched by reactive ion etching. A shadow mask defined by EBL on a 1 μm-thick poly(methyl-metacrylate)/copolymer resist bilayer is used to fabricate the flux qubit made of Al film shunting the resonator at the shorted end[30]. Before the deposition of Al film, the Nb surface is cleaned in-situ by Ar ion plasma milling for 60 s. The bolometer structure is fabricated with three three-angle deposition technique. To have a clean contact between Nb film and Al film, the Nb surface is cleaned in-situ by Ar ion plasma milling for 45 s, followed by deposition at +40° of 20 nm-thick Al lead. The Al lead is oxidized at 2.5 mbar pressure for 2 min. After that, a 3 nm-thick Al buffer layer is deposited at −6.5°, followed by deposition of 30 nm-thick Cu film at −6.5°. Finally, 90 nm-thick Al film is deposited at +20° on top of the edge of Cu film to connect

the Cu to the Nb capacitor. Finally, after liftoff in hot acetone (52° for 30 min) and cleaning with isopropyl alcohol, the substrate is cut by an automatic dicing-saw machine to the size 7 × 7 mm and wire-bonded to a copper RF-DC holder for the low-temperature characterization.

### Measurement setup

Measurements are performed in a cryogen-free dilution refrigerator at a temperature of 50 mK with the setup shown in Fig. 2d. To obtain scattering data $S_{21}$, using a VNA, a probe microwave tone is supplied to the input feedline through a 90 dB of attenuation distributed at the various temperature stages of the fridge. The output probe signal is then passed through two cryogenic circulators before being amplified first by a 40 dB cryogenic amplifier and then by a 40 dB room-temperature amplifier. The voltage $V_{th}$ is measured by a DC voltmeter through a thermocoax line. The device is mounted in a tight copper holder and covered by an Al shield.

## Data availability

Data for figures that support the manuscript are available at https://doi.org/10.5281/zenodo.15119410.

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

## Acknowledgements

We thank Mikko Möttönen, Sergey Kubatkin, Sergei Lemziakov, Vasilii Vadimov, Andrew Guthrie, Diego Subero, Dmitrii Lvov, Elias Ankerhold, and Miika Rasola for fruitful discussions and support. This work is financially supported by the Foundational Questions Institute Fund (FQXi) via Grant No. FQXi-IAF19-06, the Research Council of Finland Centre of Excellence programme grant 336810 and grant 349601 (THEPOW). We sincerely acknowledge the facilities and technical support of Otaniemi Research Infrastructure for Micro and Nanotechnologies (OtaNano) to perform this research. We thank VTT Technical Research Center for sputtered Nb films.

## Author contributions

C.D.S., Y.C.C., and J.P.P. conceived the idea of the experiment. C.D.S. designed the device with inputs from Y.C.C and R.U. C.D.S. fabricated the device, supported by J.T.P. and R.U. C.D.S. performed the experiments, supported by Y.C.C. and J.T.P. J.P.P., B.K., I.K.M. performed theoretical analysis and wrote supplementary information I, II, and IV. The data analysis was conducted by C.D.S., Y.C.C., and A.S.S. with inputs from all the authors. The manuscript was written by C.D.S. with contributions from all the authors. The figures were prepared by C.D.S. and B.K.

## Competing interests

The authors declare no competing interests.
