## [Transparent Peer Review file · Nature Communications]

Thermal spectrometer for superconducting circuits

Corresponding Author: Mr Christoforus Satrya

Version 0:

Reviewer comments:

Reviewer #1

(Remarks to the Author)

The manuscript presents a novel DC-based thermal spectrometer for superconducting circuits, demonstrated using a coplanar waveguide resonator. The authors argue that their technique surpasses conventional RF spectroscopy methods by providing a simpler measurement setup, a broader bandwidth potentially reaching 200 GHz, and a calibration-free approach in the low-power regime. Their method relies on monitoring temperature changes in an on-chip bolometer due to microwave photon absorption, allowing extraction of resonance frequency and quality factor.

The work is well-motivated and provides an alternative spectroscopic method that could be of significant interest to the community. The authors explain their experiments in detail and have performed a proof-of-concept demonstration successfully. However, some claims and comparisons raised in the manuscript require further discussion, and a few issues should be clarified.

The manuscript strongly criticizes standard RF spectroscopy methods. The authors describe RF systems as requiring complex and costly components like VNAs, low-temperature amplifiers, and circulators. While these criticisms are partially valid, they appear overstated. Even in the proposed DC measurement scheme, a microwave signal generator and input line are still required. Furthermore, challenges like grounding, parasitic resonances, and system integration remain present in both approaches, particularly in more complex setups involving qubits that have multi input lines, drive lines and flux line. Grounding, in setups with both DC measurements and flux lines, may in particular require careful grounding considerations that would not be needed for RF setups. I suggest the authors moderate their statements and acknowledge that both methods face similar challenges.

The authors use a derived formula (Equation 2) to calculate the internal quality factor. This approach relies on precise knowledge of capacitances and inductances, which may be difficult to accurately determine. Consequently, the extracted Q_i may be unreliable, particularly compared to direct extraction from standard S21 measurements where Q_i and Q_c can be directly extracted from a fit to the data. The authors should discuss the limitations of their approach and the potential uncertainties in Q_i extraction due to parameter variations.

The manuscript reports a low-photon-number internal quality factor of approximately 3000, which is relatively low for a superconducting Nb resonator. This raises a concern about whether the integration of the bolometer introduces intrinsic losses into the system. The authors should address this issue and provide additional evidence, if available, regarding the impact of the bolometer on the system's overall performance.

The measured lowest effective temperature is around 130 mK, which is somewhat higher than temperatures typically achieved in superconducting qubit measurements. This discrepancy suggests potential issues with the thermalization of the DC connection or perhaps with the DC measurements. The authors should carefully consider whether the DC wiring is adequately thermalized and discuss any potential heating effects caused by the DC components.

The inclusion of Figure 6(b) showing an S21 measurement of the third harmonic of the resonator appears redundant. The figure primarily displays noise, highlighting the limitations of RF spectroscopy in this regime. Since the main purpose of the work is to demonstrate the bolometer's capability in a high-bandwidth scenario, including this noisy RF measurement does not add value. It might be more effective to omit this sub-figure.

The comparison between the bolometer data and the S21 data in Figure 2 may also be misleading. The authors emphasize

that their DC setup provides a clearer signal compared to the S21 data, which appears hard to judge and noisy due a wiggly background that may arise from, e.g., impedance mismatches in the input/output. Additionally, due to the low Q_i/Q_c ratio, the resonator is barely visible. However, it is essential to distinguish whether the superior performance arises from the intrinsic advantages of the bolometer or from inadequacies in the RF setup. A more rigorous discussion of the underlying differences between the two setups would strengthen the manuscript. Ideally, the comparison should be between a carefully designed DC measurement setup (as done here) as compared to a carefully design RF setup (not implemented here).

This work presents a promising technique that offers a viable alternative to conventional RF spectroscopy methods. With careful consideration of the points mentioned above, the manuscript could provide a valuable contribution to the field of quantum circuit spectroscopy. The suggested revisions would strengthen the manuscript's claims and clarify the advantages of the proposed thermal spectrometer.

Reviewer #2

(Remarks to the Author)

The authors present a bolometric detector, and use it to measure a niobium superconducting resonator in parallel with a conventional microwave detection scheme. Unlike previous demonstrations of bolometric detection of superconducting quantum devices, the bolometer is fabricated on the same chip as the measured device. Using both detection methods, the authors investigate loss properties of the superconducting resonator, and show that the bolometer can measure its higher-frequency resonances without needing high-frequency readout electronics. This has the potential to greatly simplify readout schemes for superconducting qubits, particularly at high frequencies, where cryogenic electronics are costly and their performance is limited.

This work is of great interest to the superconducting quantum device community, however to be more compelling the high-frequency benefits of the device could be expanded upon, or the benefits of an on-chip bolometer given the demonstrated loss properties could made more clear. For example, using the manuscript data the authors could provide new insights by calculating the losses of the high-frequency resonator mode. Presently we find the manuscript is more suitable for a journal focusing on applications, and would not support publishing in nature communications.

In addition, we believe the following questions should be addressed to strengthen the validity of the manuscript:

1. One of the central claims of the manuscript is the use of the bolometer as an alternate method for measuring the properties of a superconducting resonator, including its linewidth. Yet the different methods of measuring the linewidth as presented in Fig. 5a and 7b significantly disagree both in magnitude and qualitative power scaling. This discrepancy should be explained, and the power dependence of the linewidth should be investigated similar to 5b. If Fig. 7b is in fact showing the linewidth of the higher-frequency harmonic, this should be clearly stated and reflected in Fig. 7a to avoid misleading the reader.
2. Based on the measured linewidth data, the internal losses of the superconducting resonator are calculated based in part on the circuit parameters. The authors should consider how the uncertainty in fabricated circuit parameters affects the reported values of internal Q: comparing propagated error bars on linewidth versus quality factor in Fig. 5 could help inform the reader whether the significant fluctuations in Q are systematic or a symptom of measurement.
3. For superconducting quantum devices, it can be important to measure physics in the single-photon regime. It would be helpful to add the average number of photons resonator as a secondary axis to the input power Fig. 5 and Fig. 7 to help readers understand which limit the resonator is in. Furthermore, could the authors comment how much background heating power would affect the bolometers ability to measure the resonator in the single-photon limit? If so, how would this be affected by the physical temperature of the sample?
4. One valuable asset of RF measurements of superconducting qubits is rapid readout, which can be done in around 20 ns. To make this measurement technique compelling specifically for qubit measurement, the authors could estimate a lower bound for the response time of the bolometer: the effective cooling rate of the bolometer itself may also be an interesting parameter.
5. In this work, the superconducting resonator is measured with both RF spectroscopy in addition to the bolometric method. It would be valuable to test the validity of both methods by comparing the internal loss rate extracted from both measurements.
6. At low power, the internal loss of the resonator is found to be limited by two-level systems with a quality factor around 10^3 , which is substantially lower than two-level system loss measured in standard microwave resonators also fabricated from niobium. An on-chip bolometer is less compelling for superconductive device readout if the co-fabrication technique results in significant loss degradation in the superconducting device. Could the authors elaborate as to where in their resonator or fabrication process this loss comes from and suggest pathways to mitigate it in future devices?

Finally a few minor notes:

1. In the abstract, the bolometer is described as "calibration-free" however multiple calibrations are described in the text. The authors should clarify whether or not these calibrations are required to know what regime the bolometer is operating in.
2. On a related note, the circuit design requires four DC-coupled wires for measuring the bolometer and calibrating it. As wire count is often an issue in larger, more complex circuits, is it feasible to eliminate some of these wires, e.g. by operating calibration-free?
3. On page 6, the calibration-free regime limit of Eq. 3, given in-line, it appears the volume Σ has been replaced by V; these should be made consistent.
4. Fig. 4 vertical scale is normalized to the on-resonance power while Fig. 6a uses the un-normalized scale, as for Fig 3a; it

would be easier to compare Figs 4 and 6 if they had consistent normalization.

Reviewer #3

(Remarks to the Author)

Dear authors,

In the manuscript "Thermal spectrometer for superconducting circuits", the authors demonstrate DC readout of a coplanar waveguide resonator by coupling an on-chip bolometer to the superconducting element. The authors argue that the proposed DC measurements scheme can ideally substitute the use of expensive RF circuitry (for instance, a vector network analyzer) as a probe for superconducting circuits' properties and possibly to readout the state of superconducting qubits. The DC readout has practical advantages in comparison to RF measurements, as it is not limited by the cables and instruments bandwidth, typically adjusted between 4 to 8 GHz for qubit applications. Overall, the device, measurement principle, and results are very clearly communicated, and the manuscript is very well written. The Supplemental Information gives valuable insight into determined technical aspects of the manuscript. Providing an alternative scheme for on-chip DC readout is an important task for the advancement of quantum technology. Although the approach is certainly interesting and technologically relevant, several important steps need to be taken to really replace readout schemes used for superconducting qubits. However, the present work is a valuable first step towards this goal. Therefore the topic is highly interesting and can reach a large audience, compatible with Nature Communications. Nevertheless, there are a few comments and questions that we feel need to be answered before a recommendation of acceptance can be made.

1 – Our first concern has to do with the speed of the readout using the DC electronics. As the authors suggest in the end of the discussion, this speed is much lower than that achievable by RF circuitry – apparently limited to "several Hz", as stated. For practical applications, a quantum computer for example, this would severely limit the speed at which operations can be performed. Can the authors provide insight on how or what if it is possible to boost the performance of the readout mechanism? In the same spirit, how long does the frequency sweep in Fig. 2 takes? What is the influence of this speed in the quality of the data obtained?

2 – As also stated by the authors, the presence of the bolometer directly and negatively affects the total quality factor of the resonator. Would any efforts to improve the readout time in turn compromise the quality factor and, therefore, the ability to distinguish between different states in a qubit?

3 – In Fig. 2f, the authors show the simultaneous readout of the resonator using the VNA. As presented, the data decisively looks worse than the one measured with the bolometer, however, calibration of the VNA before use is standard practice. Can the authors comment on the comparison of the quality factors obtained by fitting the calibrated VNA signal and those obtained using the bolometer and the presented fitting model? This could strengthen their results.

4 – Similarly, Fig. 6b shows results of a transmission readout at 20-22 GHz using the VNA. A comparison is made to establish that the thermal spectrometer can measure resonance frequencies at these ranges while the VNA could not. It is true, however, as stated by the authors, that the limitation in the VNA measurement is due to the bandwidth of the amplifiers. Thus, certainly, it would simply not be possible to measure the mode 3f₀ using the VNA in this setup. Here, we acknowledge that the point the authors make is that the thermal spectrometer eliminates the bandwidth concern, but we also feel the comparison shown in Fig. 6 is unfair.

5 – A minor comment is that the authors do not mention in the main text what is the purpose of the DC voltage source (V_h), although it is clear in the Supplemental Information it is a heater used in the context of the bolometer calibration. Nevertheless, as clearly demonstrated in Figs. 2e and 3a, the voltage across the SINIS junction used to measure the bolometer temperature decreases with increasing temperature while the bias current I_b is fixed. That is indeed the result presented in Supplemental Information III. Is this trend expected? As the temperature rises in the non-superconducting metallic element, one could expect its resistance to increase. In the same spirit, why is it possible to model the bolometer as simply a regular resistor (Fig. 1c)?

6 – In Fig. 5a, why do the authors opt to show the total photon loss rate and not the total quality factor as a function of the input power? If we extract Q_t , either from the total loss rate or using the values for Q_i , Q_b , and Q_f , we find that Q_t is not much larger than 800, with Q_i around 15000 – surely smaller than typical values achieved with superconducting resonators. What are the factors contributing to such a small quality factor? Does the presence of the normal metal itself inherently deteriorates the resonator performance? This couples back to our second remark. Moreover, can the authors comment on the errors for the extracted Q_i values? It would also be interesting to plot these graphs as a function both of P_{in} and the average photon number in the resonator.

Minor remarks:

1 – In the first page, column two, the text states that the bolometer can reach a small energy resolution. We believe the authors intend to say that it has a high energy resolution and, therefore, can measure small energies.

2 – Page six, column two. The authors write $P_b = hf_0^2/Q_t$. Did they mean to say Q_b here? Otherwise, we are confused about the expression.

Yours sincerely,

Reviewer #4

(Remarks to the Author)

I co-reviewed this manuscript with one of the reviewers who provided the listed reports. This is part of the Nature Communications initiative to facilitate training in peer review and to provide appropriate recognition for Early Career

Researchers who co-review manuscripts.

Reviewer #5

(Remarks to the Author)

Version 1:

Reviewer comments:

Reviewer #1

(Remarks to the Author)

The authors have improved the manuscript and I recommend publication

Reviewer #2

(Remarks to the Author)

The authors have responded to the various points in our review, and interestingly were able to measure the flux tuning of a qubit, making this application more compelling. However it does appear that this method is relatively slow, taking it appears more than of order 100 us ($\tau = 140$ us) to perform a measurement, at least with the bolometer described in this paper, so in this form this device would be difficult to apply to qubit measurements. However the authors do describe how this might be made faster, perhaps by an order of magnitude, so that applications to qubit measurements might be possible.

Overall we feel the responses are adequate to warrant publishing this result in nature communications.

Reviewer #3

(Remarks to the Author)

Dear authors,

In the revised version of the manuscript titled "Thermal spectrometer for superconducting circuits", the authors made efforts to improve the presentation of the results and clarify some concerns raised by the reviewers. We first would like to thank the authors for these efforts. As stated in the first review, we believe the research is an interesting application that needs further development before it is broadly technologically appealing. There are unquestionably interesting advantages of the technique, and the results are physically sound. Nevertheless, some intrinsic downsides of using the thermal readout remain (for instance, the large thermal relaxation times). We acknowledge that some of these downsides are more clearly communicated. We believe that the inclusion of the proof of principal qubit readout in the Supplemental, although with low signal-to-noise ratio, makes a valuable statement for the promise of the technique. Overall, the revision improved several aspects of the text. We believe that the results will find a large audience – and that specialized readers will be able to understand the positive and negative aspects of the proposed approach. Considering these facts, we have not changed our conclusions, and we believe that the manuscript meets the criteria for publication in Nature Communications.

Yours sincerely,

Reviewer #4

(Remarks to the Author)

Reviewer #5

(Remarks to the Author)

We thank the referees for their constructive comments. We appreciate their feedback, and the manuscript is much improved as a result.

We hope that our following responses, together with the revised manuscript, can address the comments of the referees. Additionally, we have attached a pdf showing the differences between the original and the revised manuscript for clarity.

Referee 1:

The manuscript presents a novel DC-based thermal spectrometer for superconducting circuits, demonstrated using a coplanar waveguide resonator. The authors argue that their technique surpasses conventional RF spectroscopy methods by providing a simpler measurement setup, a broader bandwidth potentially reaching 200 GHz, and a calibration-free approach in the low-power regime. Their method relies on monitoring temperature changes in an on-chip bolometer due to microwave photon absorption, allowing extraction of resonance frequency and quality factor

The work is well-motivated and provides an alternative spectroscopic method that could be of significant interest to the community. The authors explain their experiments in detail and have performed a proof-of-concept demonstration successfully. However, some claims and comparisons raised in the manuscript require further discussion, and a few issues should be clarified.

We acknowledge the referee for his/her thoughtful feedback and for recognizing the motivation and significance of our work.

Q1.1: The manuscript strongly criticizes standard RF spectroscopy methods. The authors describe RF systems as requiring complex and costly components like VNAs, low-temperature amplifiers, and circulators. While these criticisms are partially valid, they appear overstated. Even in the proposed DC measurement scheme, a microwave signal generator and input line are still required. Furthermore, challenges like grounding, parasitic resonances, and system integration remain present in both approaches, particularly in more complex setups involving qubits that have multi input lines, drive lines and flux line. Grounding, in setups with both DC measurements and flux lines, may in particular require careful grounding considerations that would not be needed for RF setups. I suggest the authors moderate their statements and acknowledge that both methods face similar challenges.

A1.1: We thank the referee for the suggestion. We acknowledge that some of our claims are overstatements; therefore we have moderated wording and sentences in the abstract and in the main text of the revised manuscript.

Challenges like grounding and parasitic resonances remain present in both methods. With the DC spectroscopy setup, they might be partially reduced since we do not need to use an output line that consists of a chain of active and passive elements. Although the integration of DC lines in the chip can also lead to ground plane interruptions which can cause grounding and parasitic resonance problems. There are potential ways to connect the discontinuous ground planes by air-bridges [<https://doi.org/10.1063/5.0103165>, <https://doi.org/10.1116/1.4798399>], flip-chip schemes [<https://doi.org/10.1088/2058-9565/ac734b>] and parallel-plate capacitors

[<https://doi.org/10.1103/PhysRevResearch.6.L012035>]. In our device, we tried to solve this problem by having many air bonds in the chip to make the ground plane continuous.

Q1.2: The authors use a derived formula (Equation 2) to calculate the internal quality factor. This approach relies on precise knowledge of capacitances and inductances, which may be difficult to accurately determine. Consequently, the extracted Q_i may be unreliable, particularly compared to direct extraction from standard S21 measurements where Q_i and Q_c can be directly extracted from a fit to the data. The authors should discuss the limitations of their approach and the potential uncertainties in Q_i extraction due to parameter variations.

A1.2: There are 2 parameters in Eq. (2) that cannot be extracted from measurements, namely C_b and C_f . They rely on FEM simulation that gives around 2% uncertainty for each of them arising mainly from model's uncertainties, which can be reduced by careful sample design. These errors are included to Fig.5b as additional error due to uncertainty in the simulation. See response **A2.2** on possible errors in the simulations.

The equivalent lumped L and C can be determined from Eqs. (2) and (3) of Supplementary Information I, since the resonance frequency f_0 can be measured. The resistance R_b can be determined by a four-probe IV measurement.

We added few sentences about these points in the main text in the revised manuscript.

Q1.3: The manuscript reports a low-photon-number internal quality factor of approximately 3000, which is relatively low for a superconducting Nb resonator. This raises a concern about whether the integration of the bolometer introduces intrinsic losses into the system. The authors should address this issue and provide additional evidence, if available, regarding the impact of the bolometer on the system's overall performance.

A1.3: In this experiment the bolometer is not included as an internal loss source. Instead, it acts as a port with external quality factor Q_b and impedance R_b . The internal quality factor Q_i is attributed only to the material and quasiparticle losses inside the resonator.

Insertion of the bolometer to the chip could introduce additional internal losses. They are associated with fabrication residuals, such as resist remaining after NIS bolometer lift-off step. Reduction of Q_i by lift-off residues was demonstrated in [<https://doi.org/10.1063/1.4993577>]. This could be mitigated by more advanced process of cleaning the lift-off residuals, such as oxygen plasma cleaning.

The measured Nb resonator is on top of AlOx/Si. The main reason of low internal quality factor Q_i in our device is this AlOx sublayer that acts as a stop-layer during RIE Nb etching. This fabrication approach provides stopping Si over-etch, so the Si surface is not damaged during Nb etching. In future, it is possible to remove this AlOx sublayer.

AlOx is known to be a source of TLSs and it lowers Q_i in our device. The figure below shows a measurement of Q_i vs power of a typical bare Nb resonator (without the presence of bolometer). The left one is a Nb resonator on AlOx/Si and the right one is Nb resonator on Si. It shows that with the AlOx sublayer Q_i is suppressed to around ~ 10000 at low power.

Measurement of Q_i vs power of a typical bare Nb resonator. The left is Nb resonator on AlOx/Si and the right is Nb resonator on Si [data are from *Sergei Lemziakov, Dmitrii Lvov, Elias Ankerhold*].

Another possible additional source of low Q_i is residual oxide in the interface between Nb and Al in the shorted end of the resonator. In the fabrication process we tried to clean the Nb surface with Ar plasma before depositing Al film on Nb. But there could still be some impurities in this interface. A similar case is studied in [<https://doi.org/10.1063/1.4939299>], where they found that Nb resonator connected to Al bridge results to $Q_i \sim 5000$ when the interface was not cleaned.

Q1.4: The measured lowest effective temperature is around 130 mK, which is somewhat higher than temperatures typically achieved in superconducting qubit measurements. This discrepancy suggests potential issues with the thermalization of the DC connection or perhaps with the DC measurements. The authors should carefully consider whether the DC wiring is adequately thermalized and discuss any potential heating effects caused by the DC components.

A1.4: The measured lowest bolometer temperature is $T_b = 130$ mK, which is due to background power $P_e \sim 2$ fW (see Supplementary Information III). This background-power flow indeed mostly comes from DC lines that connect to the NIS junctions [<https://www.nature.com/articles/nature05276>]. We added this information in the main text.

The measured bolometer temperature T_b is local temperature of Cu film resistor and not the sample substrate, which depends also on volume and electron-phonon coupling constant $T_b = \sqrt[5]{T_0^5 + \frac{P_e}{\Sigma\Omega}}$ [<https://doi.org/10.1103/RevModPhys.78.217>]. The temperature of the substrate is expected to be around the base temperature of 50 mK, which could be higher for example due to bad thermalization between the chip and the holder.

Q1.5: The inclusion of Figure 6(b) showing an S21 measurement of the third harmonic of the resonator appears redundant. The figure primarily displays noise, highlighting the limitations of RF spectroscopy in this regime. Since the main purpose of the work is to demonstrate the bolometer's capability in a high-bandwidth scenario, including this noisy RF measurement does not add value. It might be more effective to omit this sub-figure.

A1.5: Thanks for the feedback. We omitted Fig. 6b.

Q1.6: The comparison between the bolometer data and the S21 data in Figure 2 may also be misleading. The authors emphasize that their DC setup provides a clearer signal compared to the S21 data, which appears hard to judge and noisy due to a wiggly background that may arise from, e.g., impedance mismatches in the input/output. Additionally, due to the low Q_i/Q_c ratio, the resonator is barely visible. However, it is essential to distinguish whether the superior performance arises from the intrinsic advantages of the bolometer or from inadequacies in the RF setup. A more rigorous discussion of the underlying differences between the two setups would strengthen the manuscript. Ideally, the comparison should be between a carefully designed DC measurement setup (as done here) as compared to a carefully design RF setup (not implemented here)“

A1.6: As it was mentioned in response **A1.3**, bolometer can be interpreted as a Port-3 with impedance R_b and external quality factor Q_b . In contrast to RF output Port-2, it does not measure field quadrature, but power flow to the port, losing information about phase. So, further comparison will be devoted to $|S21|$ notch-type measurement and $|S31|$ transmission type measurement.

There are two major differences between these measurement types, namely coupling regime and background information.

To compare coupling regimes let us introduce the effective $Q_i^* \sim 1600$ of $|S21|$ measurement: $1/Q_i^* = 1/Q_i + 1/Q_b$; and effective $Q_c^* \sim 800$ of $|S31|$ measurement: $1/Q_c^* = 1/Q_b + 1/Q_f$. The $|S21|$ measurement is performed in intermediate coupling regime, when $Q_i^* \sim Q_f$, that is not good in terms of SNR (especially in single-photon regime) since the resonator depth reduces to $< \sim 3$ dB. In turn, $|S31|$ measurement is carried out in over-coupled regime, when $Q_c^* \ll Q_i \sim 10^3$, which provides better SNR with respect to $|S21|$ measurement with the same port parameters.

The other major difference between the two types of measurements is the background signal. The $|S21|$ acquired with a notch-type resonator keeps that background information. This allows to get Q_i^* and Q_f values directly from the S21 curve fit. For resonator with low loaded (total) quality factor Q_t , whose shape is affected by background features, such measurement can give results affected by systematic errors. In turn, transmission-type $|S31|$ measurement loses the information about background, which allows to get more stable results, but allows to measure only Q_t , that requires additional input power calibration or circuit simulations (see Eq. (2) and Supplementary Information I) to evaluate Q_i , Q_b and Q_f .

For more information about background and SNR see [<https://doi.org/10.1063/1.4907935>], [<https://doi.org/10.1109/TASC.2016.2625767>].

The background problem of $|S21|$ measurement can be potentially solved by subtracting background shape. The background can be measured at high temperatures to suppress resonator quality factor by thermal quasiparticles. See also responses **A2.5** and **A3.3**.

This work presents a promising technique that offers a viable alternative to conventional RF spectroscopy methods. With careful consideration of the points mentioned above, the manuscript could provide a valuable contribution to the field of quantum circuit

spectroscopy. The suggested revisions would strengthen the manuscript's claims and clarify the advantages of the proposed thermal spectrometer.

We thank the referee for the positive assessment of our work.

Referee 2:

The authors present a bolometric detector, and use it to measure a niobium superconducting resonator in parallel with a conventional microwave detection scheme. Unlike previous demonstrations of bolometric detection of superconducting quantum devices, the bolometer is fabricated on the same chip as the measured device. Using both detection methods, the authors investigate loss properties of the superconducting resonator, and show that the bolometer can measure its higher-frequency resonances without needing high-frequency readout electronics. This has the potential to greatly simplify readout schemes for superconducting qubits, particularly at high frequencies, where cryogenic electronics are costly and their performance is limited.

This work is of great interest to the superconducting quantum device community, however to be more compelling the high-frequency benefits of the device could be expanded upon, or the benefits of an on-chip bolometer given the demonstrated loss properties could be made more clear. For example, using the manuscript data the authors could provide new insights by calculating the losses of the high-frequency resonator mode. Presently we find the manuscript is more suitable for a journal focusing on applications, and would not support publishing in nature communications.

We thank the referee for reading the manuscript, and for the comments and feedback.

Q2.1: One of the central claims of the manuscript is the use of the bolometer as an alternate method for measuring the properties of a superconducting resonator, including its linewidth. Yet the different methods of measuring the linewidth as presented in Fig. 5a and 7b significantly disagree both in magnitude and qualitative power scaling. This discrepancy should be explained, and the power dependence of the linewidth should be investigated similar to 5b. If Fig. 7b is in fact showing the linewidth of the higher-frequency harmonic, this should be clearly stated and reflected in Fig. 7a to avoid misleading the reader.

A2.1: The presented data in Fig.7b is from a different set of measurements not from the same set of data presented in Fig.5a. We apologize for this confusion. For consistency and clarity, we decided to use the same set of data from Fig.5 for the updated Fig.7.

In Fig. 7a we showed that at low P_{in} the measured SINIS voltage V_{th} follows Lorentzian spectrum. As we described in the main text and also explained in reference [<https://doi.org/10.1103/RevModPhys.93.041001>], this is because the temperature raise is small thus the $P_b \sim V_{th}$. In this regime, that we called 'calibration-free regime', the linewidth can be obtained directly from the measured data V_{th} , without the need to convert it to P_b . In Fig. 7b we tried to prove this point by comparing linewidth extracted from V_{th} and P_b . At low powers the extracted linewidths are approximately the same justifying the calibration-free regime, and they are deviating toward high powers due to large temperature rise.

Q2.2: Based on the measured linewidth data, the internal losses of the superconducting resonator are calculated based in part on the circuit parameters. The authors should consider how the uncertainty in fabricated circuit parameters affects the reported values of internal Q : comparing propagated error bars on linewidth versus quality factor in Fig. 5 could help inform the reader whether the significant fluctuations in Q are systematic or a symptom of measurement.

A2.2: Thanks for the feedback. We added error bars due fitting uncertainty in Figs. 5 and 7b. Also, we added in Fig. 5b additional error estimates due to uncertainty in the simulated circuit parameters.

Response to this is related to responses to the first referee **A1.2**

Among all parameters used in Eq. (2), only C_b and C_f rely on design values that come from finite-element method (FEM) simulations. We know quite precisely the dimensions of the capacitors/couplers from the SEM characterization. Here, we assume that the main source of systematic errors come from the FEM simulation. All simulations here are performed in COMSOL Multiphysics software.

The first issue is mesh convergence. Improper mesh density can lead to systematic errors. This problem is solved via adaptive mesh algorithm (see example below), leading to errors 0.1-1% depending on number of elements used during simulation. The main limitation of this error is the available RAM. For desktop PC with 8Gb free memory one can reach precision of 0.1% in a reasonable time, but for heavier models larger RAM consumption is required. The second one is the bounding box of the model. Here the bounding box that is $200\ \mu\text{m}$ away from the structure is used, providing 0.1% error with the same RAM restrictions. The final problem is the model structure. The CPW coupler simulation can be done via 2D cross-section, that allows to consider thin AlOx sublayer, the thickness of Nb, and the trapezoid shape of the edge. Nevertheless, the additional coupling in the resonator rounding introduces an additional error of 2%, that becomes a bottleneck of the coupler simulation. In turn, the 3D model of the interdigital capacitor cannot take into account these effects with reasonable RAM consumption. Also, it does not include shape deviations caused by EBL proximity effect.

The overall relative errors of simulation are assumed to be 2% for both C_b and C_f , that is now included in Fig. 5b.

Example of mesh convergence algorithm result

Field distribution and mesh of interdigital capacitor simulation, the last step of mesh refinement algorithm

Top and side view of Nb edge, where trapezoid shape pops up.

EBL proximity effect that deviates the shape of the interdigital capacitor's lead

Q2.3: For superconducting quantum devices, it can be important to measure physics in the single-photon regime. It would be helpful to add the average number of photons resonator as a secondary axis to the input power Fig. 5 and Fig. 7 to help readers understand which limit the resonator is in. Furthermore, could the authors comment how much background heating power would affect the bolometers ability to measure the resonator in the single-photon limit? If so, how would this be affected by the physical temperature of the sample?

A2.3: We added average photon number of resonator N on the top y-axis in Fig. 3b, Fig. 5 and Fig. 7b. We created and added ‘Supplementary Information IV’ which describes how we derive the average photon number N .

The background heating power P_e will affect the saturation temperature of the bolometer T_b . In our device, $P_e \sim 2$ fW corresponds to temperature saturation $T_b \sim 130$ mK. And this gives the noise-equivalent power (NEP) due to thermal fluctuations as $NEP_{th} = \sqrt{4k_B T_b^2 G_{th}} = 2.6 \times 10^{-19} \text{ W}/\sqrt{\text{Hz}}$. While single photon power is $P_b = hf_0^2 2\pi/Q_b = 11.9 \times 10^{-17} \text{ W}$. So it is possible to measure single photon with 1 Hz bandwidth in this NEP_{th} condition. However, if for example background heating is bigger and results to saturation $T_b \sim 600$ mK, the noise level becomes $NEP_{th} \sim 2.5 \times 10^{-17} \text{ W}/\sqrt{\text{Hz}}$. In this condition, it would be more difficult to see the single photon signal since the noise level is already in the same order as the signal.

In addition, we added Supplementary Information V showing preliminary results of thermal spectroscopy of resonator-qubit measured at the low photon regime ($N \sim 0.6$).

Q2.4: One valuable asset of RF measurements of superconducting qubits is rapid readout, which can be done in around 20 ns. To make this measurement technique compelling specifically for qubit measurement, the authors could estimate a lower bound for the response time of the bolometer: the effective cooling rate of the bolometer itself may also be an interesting parameter.

A2.4: In the presented paper, the bolometric measurement is done by low-bandwidth voltage amplifier for spectroscopy measurements in a few Hz bandwidth. It is possible to operate the bolometer in a fast readout, but this requires a different setup. In this reference [<https://doi.org/10.1103/PhysRevApplied.3.014007>], to enable a fast readout to achieve 1 μs time resolution, the bolometer was operated by embedding the NIS junction in a LC resonant circuit with intermediate frequency RF setups (~ 625 MHz) for probing. Here the response time of the bolometer is 1-10 μs .

The effective cooling rate of the bolometer/thermal relaxation rate in our device is $\tau = C_h/G_{th}$, where C_h is electronic heat capacity of the Cu and G_{th} is thermal conductance to phonon bath, where $C = (71 \text{ J K}^{-2} \text{ m}^{-3})\Omega T_b$ and $G_{th} = 5\Sigma\Omega T_0^4$ [<https://doi.org/10.1103/PhysRevApplied.3.014007>]. With $T_b = 130$ mK, we estimate $\tau = 148 \mu\text{s}$ at $T_0 = 50$ mK.

Q2.5: In this work, the superconducting resonator is measured with both RF spectroscopy in addition to the bolometric method. It would be valuable to test the validity of both methods by comparing the internal loss rate extracted from both measurements.

A2.5: The answer to this question is related to response **A1.6** and **A3.3**.

The extraction of all quality factors (Q_i , Q_b , Q_f) from the measurement requires combination of fit results of $|S_{21}|$ measurement, that gives Q_f and Q_i^* , and $|S_{31}|$ measurements, that give Q_i and Q_c^* . Unfortunately, $|S_{31}|$ is a transmission-type measurement, that loses information about background, so an input power flow calibration is required [<https://doi.org/10.1063/1.4907935>]. An alternative way, employed here, is to use a circuit model to extract Q_f and Q_b .

Below one can see fit results of $|S_{21}|$ (orange) and $|S_{31}|$ (blue) measurement. Error bars correspond to the fit error. The $|S_{21}|$ curve fit result is affected by background shape and gives unreliable results, that depends on the background model and dataset range. Due to this fact, it is not included in the work.

Extracted frequency shift and total/loading quality factor vs P_{in} . Blue are from bolometer (S31) measurement and orange are from S21 measurement.

Q2.6: At low power, the internal loss of the resonator is found to be limited by two-level systems with a quality factor around 10^3 , which is substantially lower than two-level system loss measured in standard microwave resonators also fabricated from niobium. An on-chip bolometer is less compelling for superconductive device readout if the co-fabrication technique results in significant loss degradation in the superconducting device. Could the authors elaborate as to where in their resonator or fabrication process this loss comes from and suggest pathways to mitigate it in future devices?

A2.6: The measured Nb resonator is on top of AlOx/Si. The main reason of low Q_i in our device is this AlOx sublayer that acts as a stop-layer during RIE Nb etching. See response to the first referee **A1. 3**. In the future we can remove this AlOx layer.

Q2.7: In the abstract, the bolometer is described as “calibration-free” however multiple calibrations are described in the text. The authors should clarify whether or not these calibrations are required to know what regime the bolometer is operating in.

A2.7: In Fig.7a we showed that at low P_{in} the measured SINIS voltage V_{th} follows Lorentzian spectrum. As we described in the main text and also explained in reference [<https://doi.org/10.1103/RevModPhys.93.041001>], this is because the temperature rise is small thus $P_b \sim$ change in V_{th} . In this regime, what we called ‘calibration-free regime’, the linewidth can be obtained directly from the measured V_{th} , without a need to convert it to P_b . In Fig.7b we prove this point by comparing linewidth extracted from V_{th} and P_b . At low powers the extracted linewidths are approximately equal and they are deviating toward high powers due to large temperature rise. Thus, if the bolometer operates in the regime where the temperature raise is small, it is not necessary to convert V_{th} to P_b .

We added additional description in the main text on the calibration-free regime of the bolometer.

Q2.8: On a related note, the circuit design requires four DC-coupled wires for measuring the bolometer and calibrating it. As wire count is often an issue in larger, more complex circuits, is it feasible to eliminate some of these wires, e.g. by operating calibration-free?

A2.8: Yes, it is possible. In principle the calibration of V_{th} against temperature, as shown in Fig.S3c of Supplementary Information III, does not require the heater junction V_h , which we use here to test the thermometry response. In addition, it is also possible to use a single NIS junction as a thermometer [<https://doi.org/10.1103/PhysRevApplied.4.034001>]. In principle it is possible to operate the bolometer only by one NIS junction.

Q2.9: On page 6, the calibration-free regime limit of Eq. 3, given in-line, it appears the volume Σ has been replaced by V ; these should be made consistent.

A2.9: Thank you for notification. We fixed this.

Q2.10: Fig. 4 vertical scale is normalized to the on-resonance power while Fig. 6a uses the un-normalized scale, as for Fig 3a; it would be easier to compare Figs 4 and 6 if they had consistent normalization.

A2.10: Thanks for noting this. We now changed the style of Fig. 6 to be consistent with that of Fig. 4.

Referee 3:

In the manuscript “Thermal spectrometer for superconducting circuits”, the authors demonstrate DC readout of a coplanar waveguide resonator by coupling an on-chip bolometer to the superconducting element. The authors argue that the proposed DC measurements scheme can ideally substitute the use of expensive RF circuitry (for instance, a vector network analyzer) as a probe for superconducting circuits’ properties and possibly to readout the state of superconducting qubits. The DC readout has practical advantages in comparison to RF measurements, as it is not limited by the cables and instruments bandwidth, typically

adjusted between 4 to 8 GHz for qubit applications. Overall, the device, measurement principle, and results are very clearly communicated, and the manuscript is very well written. The Supplemental Information gives valuable insight into determined technical aspects of the manuscript. Providing an alternative scheme for on-chip DC readout is an important task for the advancement of quantum technology. Although the approach is certainly interesting and technologically relevant, several important steps need to be taken to really replace readout schemes used for superconducting qubits. However, the present work is a valuable first step towards this goal. Therefore the topic is highly interesting and can reach a large audience, compatible with Nature Communications. Nevertheless, there are a few comments and questions that we feel need to be answered before a recommendation of acceptance can be made.

We thank the referee for his/her positive assessment and thoughtful feedback. We appreciate his/her recognition of our work's clarity, significance, and potential impact

Q3.1: Our first concern has to do with the speed of the readout using the DC electronics. As the authors suggest in the end of the discussion, this speed is much lower than that achievable by RF circuitry – apparently limited to “several Hz”, as stated. For practical applications, a quantum computer for example, this would severely limit the speed at which operations can be performed. Can the authors provide insight on how or what if it is possible to boost the performance of the readout mechanism? In the same spirit, how long does the frequency sweep in Fig. 2 takes? What is the influence of this speed in the quality of the data obtained?

A3.1: In the presented paper, the bolometric measurement is done by low-bandwidth voltage amplifier for spectroscopy measurements in a few Hz bandwidth. It is possible to operate the bolometer in a fast readout, but this requires a different setup. In this reference [<https://doi.org/10.1103/PhysRevApplied.3.014007>], to enable a fast readout to achieve 1 μ s time resolution, we operated the bolometer by embedding the NIS junction in a LC resonant circuit with intermediate frequency RF setups (~ 625 MHz) for probing. We performed time-resolved measurements and could measure the thermal relaxation rates of bolometer to be $\tau \sim 10^2 \mu$ s at 50 mK bath temperature.

In Fig. 2e, it took 0.2 seconds to measure each point (20 ms integration time x 10 averaging) and for the whole frequency sweep the total time is 120.02 s. If we try to reduce the total time for example by reducing the integration time or averaging the data quality (signal-to-noise ratio) will be worse.

Q3.2: As also stated by the authors, the presence of the bolometer directly and negatively affects the total quality factor of the resonator. Would any efforts to improve the readout time in turn compromise the quality factor and, therefore, the ability to distinguish between different states in a qubit?

A3.2: In this experiment the bolometer is not attributed as internal loss source. Instead, it acts as a port with external quality factor Q_b and impedance R_b . The internal quality factor Q_i is attributed only to the material and quasiparticle losses inside the resonator. If we increase external quality factors Q_f and Q_b than the total quality factor Q_t will be also higher.

Readout time is proportional to $1/\gamma_c$, where $\gamma_c = f_0 \cdot (1/Q_b + 1/Q_f)$. Distinguishability of states depends on $\kappa = f_0 \cdot (1/Q_b + 1/Q_f + 1/Q_i) \sim \gamma_c$. So, yes, readout time improvement breaks qubit state distinguishability.

In principle, potential qubit dispersive readout with such a scheme is similar to the one that is used in cQED experiments with transmission-type resonator [<https://doi.org/10.1109/TASC.2005.850084>][<https://doi.org/10.1103/PhysRevLett.95.060501>], but with the bolometer used as a port that can measure $|S_{31}|$ signal.

Q3.3: In Fig. 2f, the authors show the simultaneous readout of the resonator using the VNA. As presented, the data decisively looks worse than the one measured with the bolometer, however, calibration of the VNA before use is standard practice. Can the authors comment on the comparison of the quality factors obtained by fitting the calibrated VNA signal and those obtained using the bolometer and the presented fitting model? This could strengthen their results.

A3.3: Response for this is included to response **A1.6 and A2.5**.

As far as we understand, calibration of VNA means measurement of a feedline background at 4K, when resonator quality factor is suppressed by quasiparticles. Unfortunately, such measurements were not done. Few attempts to fit the $|S_{21}|$ signal were performed in order to extract Q_f and Q_i from S_{21} measurements. In these attempts background shape (linear part with ripples) has been included to the fit procedure. The result is shown in **A1.6 and A2.5**. Such fit approach gives a similar result (at least mean value) as the bolometer's data fit but depends on the data range and fit function, thus being unreliable.

Q3.4: Similarly, Fig. 6b shows results of a transmission readout at 20-22 GHz using the VNA. A comparison is made to establish that the thermal spectrometer can measure resonance frequencies at these ranges while the VNA could not. It is true, however, as stated by the authors, that the limitation in the VNA measurement is due to the bandwidth of the amplifiers. Thus, certainly, it would simply not be possible to measure the mode $3f_0$ using the VNA in this setup. Here, we acknowledge that the point the authors make is that the thermal spectrometer eliminates the bandwidth concern, but we also feel the comparison shown in Fig. 6 is unfair.

A3.4: Thanks for the feedback. We omitted Fig. 6b.

Q3.5: A minor comment is that the authors do not mention in the main text what is the purpose of the DC voltage source (V_h), although it is clear in the Supplemental Information it is a heater used in the context of the bolometer calibration. Nevertheless, as clearly demonstrated in Figs. 2e and 3a, the voltage across the SINIS junction used to measure the bolometer temperature decreases with increasing temperature while the bias current I_b is fixed. That is indeed the result presented in Supplemental Information III. Is this trend expected? As the temperature rises in the non-superconducting metallic element, one could expect its resistance to increase. In the same spirit, why is it possible to model the bolometer as simply a regular resistor (Fig. 1c)?

A3.5: We added this description of V_h in the main text.

At temperature range 50-400 mK of our measurement, the resistance of the Cu film is constant and saturates already due to defects and impurities [<https://www.nature.com/articles/srep10705>]. Also, the conductance is dominated by the SINIS junctions, since resistance of the SINIS is in $\sim k\Omega$ range while Cu is in $\sim \Omega$ range. The measured voltage V_{th} drops are due to temperature changes in thermal Fermi-Dirac distribution of electrons in Cu, which is the principle of NIS thermometry [<https://doi.org/10.1103/RevModPhys.78.217>].

Q3.6: In Fig. 5a, why do the authors opt to show the total photon loss rate and not the total quality factor as a function of the input power? If we extract Q_t , either from the total loss rate or using the values for Q_i , Q_b , and Q_f , we find that Q_t is not much larger than 800, with Q_i around 15000 – surely smaller than typical values achieved with superconducting resonators. What are the factors contributing to such a small quality factor? Does the presence of the normal metal itself inherently deteriorates the resonator performance? This couples back to our second remark. Moreover, can the authors comment on the errors for the extracted Q_i values? It would also be interesting to plot these graphs as a function both of P_{in} and the average photon number in the resonator.

A3.6: We added the plot of Q_t in Fig.5a.

The bolometer acts as a port with external quality factor Q_b and impedance R_b . The internal quality factor Q_i is attributed only to the material and quasiparticle losses inside the resonator. The low Q_i in our device mostly is attributed to AlOx sublayer. In the future we can remove this AlOx layer. See responses to the first and second referee **A1. 3 and A2.6**.

Total quality factor Q_t is around ~ 700 because in this device we have relatively low external quality factors $Q_f \sim 1600$ and $Q_b \sim 1700$. We can in the future increase Q_t , by designing the device with higher external quality factors Q_f and Q_b .

Comments on the error in the extracted Q_i can be find in responses **A1.2** and **A2.2**.

We added average photon number N at top y-axis in Fig. 3b, Fig. 5(a-b), and Fig. 7b. We created and added ‘Supplementary Information IV’ which describes how to derive the average photon number N .

Q3.7: In the first page, column two, the text states that the bolometer can reach a small energy resolution. We believe the authors intend to say that it has a high energy resolution and, therefore, can measure small energies.

A3.7: Thanks for noticing this. We changed this.

Q3.8: Page six, column two. The authors write $P_b = hf_0^2/Q_t$. Did they mean to say Q_b here? Otherwise, we are confused about the expression.

A3.8: Thanks for noticing this. Yes, the correct form is $P_b = hf_0^2 2\pi/Q_b$. We updated this in the revised manuscript. In addition, we created Supplementary Information IV describing the derivation of P_b and N .

Referee 4:

We thank the referee for reading the manuscript, and for the comments and feedback.

Referee 5:

We thank the referee for reading the manuscript, and for the comments and feedback.

Reviewer #1

The authors have improved the manuscript and I recommend publication

We thank the referee for reading our responses and the revised manuscript. We are grateful for the reviewer for recommending to publish our work in Nature Communications.

Reviewer #2

The authors have responded to the various points in our review, and interestingly were able to measure the flux tuning of a qubit, making this application more compelling. However it does appear that this method is relatively slow, taking it appears more than of order 100 us ($\tau = 140$ us) to perform a measurement, at least with the bolometer described in this paper, so in this form this device would be difficult to apply to qubit measurements. However the authors do describe how this might be made faster, perhaps by an order of magnitude, so that applications to qubit measurements might be possible.

Overall we feel the responses are adequate to warrant publishing this result in nature communications.

We thank the referee for reading our responses and the revised manuscript, and for the recommendation to publish our work in Nature Communications.

Reviewer #3

Dear authors,

In the revised version of the manuscript titled “Thermal spectrometer for superconducting circuits”, the authors made efforts to improve the presentation of the results and clarify some concerns raised by the reviewers. We first would like to thank the authors for these efforts. As stated in the first review, we believe the research is an interesting application that needs further development before it is broadly technologically appealing. There are unquestionably interesting advantages of the technique, and the results are physically sound. Nevertheless, some intrinsic downsides of using the thermal readout remain (for instance, the large thermal relaxation times). We acknowledge that some of these downsides are more clearly communicated. We believe that the inclusion of the proof of principal qubit readout in the Supplemental, although with low signal-to-noise ratio, makes a valuable statement for the promise of the technique. Overall, the revision improved several aspects of the text. We believe that the results will find a large audience – and that specialized readers will be able to understand the positive and negative aspects of the proposed approach. Considering these facts, we have not changed our conclusions, and we believe that the manuscript meets the criteria for publication in Nature Communications.

Yours sincerely,

We thank the referee for recognizing our work and for recommending its publication in Nature Communications.

Regarding the large thermal relaxation times, for the future device, this can be improved by using an absorber with a lower heat capacity, for example one made of graphene.

Reviewer #4

We thank and appreciate the referee for reading our responses and revised manuscript.

Reviewer #5

We thank and appreciate the referee for reading our responses and revised manuscript.